# Surface Area of Wood Influences the Effects of Fungal Interspecific Interaction on Wood Decomposition—A Case Study Based on *Pinus densiflora* and Selected White Rot Fungi

**DOI:** 10.3390/jof8050517

**Published:** 2022-05-18

**Authors:** Yu Fukasawa, Koji Kaga

**Affiliations:** Graduate School of Agricultural Science, Tohoku University, 232-3 Yomogida, Naruko, Osaki 989-6711, Miyagi, Japan; kagamoku@gmail.com

**Keywords:** basidiomycetes, fungal interaction, lignin, surface–volume geometry, white rot fungi, wood decomposition

## Abstract

Wood decomposer basidiomycetes are the major agents of lignocellulose decomposition in dead wood. As their interspecific interaction affects wood decomposition, difference in interaction area may alter the magnitude of the effects. This study examines the effects of wood surface area on decomposition by interacting basidiomycetes using laboratory incubation experiments with pine sapwood as a model. Two types of pine wood blocks with equal volume but identical surface area were prepared for colonization by one of four white rot basidiomycete species. The colonized wood blocks were then placed on agar media already colonized by the same strain or one of the other species, simulating fungal monoculture and interspecific interactions on wood surface. Results demonstrated that the decay rate of wood was greater in wood with larger surface, and wood decay was accelerated by the interaction of two fungal species in wood with larger surface but not in wood with smaller surface. In contrast, lignin decomposition was influenced by the competitor in wood with smaller surface but not in wood with larger surface. These results suggest that the observed promotion of decay by fungal interspecific interaction might not be attributable to the resource partitioning between fungal species but to the accelerated carbon of competition cost compensation in this case.

## 1. Introduction

Dead wood represents a large carbon stock in terrestrial ecosystems and thus plays an important role in global carbon cycling [1,2]. Wood decomposer basidiomycetes are the major agents of lignocellulose decomposition in dead wood [3,4]. The effects of basidiomycete colonization on wood decomposition are essential and the largest driving factor even at a continental spatial scale, in which wood decomposition was previously believed to be regulated by global factors such as macroclimate [5]. Understanding the relationships across basidiomycete community dynamics, biotic interactions and wood decomposition is crucial for modeling and predicting ecosystem carbon cycling [6,7].

As a dead wood hosts many fungal species [8,9,10], numerous interspecific interactions might exist within dead wood. Even in a small dead wood occupied by a single fungus, species interactions must occur between the fungus inside the wood and the fungi in the outer environment, e.g., soil. Competition for wood volume (space) is the predominant type of fungal interaction determining the community development of wood decomposer basidiomycetes [11,12]. When two fungal colonies meet, combat (antagonism) occurs between them to defend their territory or acquire more territory. The outcomes of combative interactions can be replacement, where one fungus gains territory from another; deadlock, when neither fungus obtains additional territory; and reciprocal replacement, where one fungus gains territory from the other in one region, but loses it elsewhere [11,12]. In the interaction front of the two colonies, a variety of antagonistic reactions could occur [4,12], such as the production of secondary metabolites (pigments and volatiles), enzyme production, and the formation of sclerotal, dense mycelial plates appearing as demarcation lines. Therefore, fungal interspecific interactions are energetically expensive and thus could influence the decay activities of fungi. The impact could be positive (accelerate decomposition) if the exhaustion of energy accelerates carbon acquisition by decomposition (competition cost hypothesis) or the competing species utilize complementary wood components (complementary hypothesis). The effects could be negative (retard decomposition) if energy allocation to competition reduces energy allocation to decay enzyme production (energy tradeoff hypothesis). Previous studies have reported positive [13,14,15,16] and negative [14,17,18,19,20] effects of fungal species interactions on wood decomposition. Nevertheless, what determines the positive and negative effects of fungal interspecific interactions on wood decay is poorly understood. Fukasawa et al. [20] shed light on this argument by demonstrating that wood decomposition is reduced at the interaction front compared to that at the rear of the fungal colonies, and smaller wood decays faster than larger wood during fungal interaction. These results suggest that energy cost for competition might promote wood decomposition except at the interaction front where energy allocation to antagonistic responses reduce is reduced to decay enzyme production. This leads to a new hypothesis that interaction area is important for determining the effects of fungal interspecific interactions on wood decomposition. If fungal interspecific interaction occurs on wood surface, wood with larger surface might exhibit a greater effect than wood with smaller surface, and the effect might be negative due to enlarged energy allocation to antagonistic responses.

Fukasawa described a decomposition process of *Pinus densiflora* (Siebold & Zucc.) dead wood dominated by white rot fungi that decay wood lignin preferentially or at the same rate as that of holocellulose in field conditions [21]. In *P. densiflora* dead wood, Fukasawa found a simultaneous decay of lignin and holocellulose at the early stages of wood decomposition and a selective decomposition of holocellulose at the late stages, resulting in an increase of lignin content at the late stages [21]. Despite the widespread assumption that the dead wood of conifer trees is dominated by brown rot fungi that decay wood holocellulose preferentially with little modification of lignin, white rot of coniferous dead wood is quite common [22,23,24,25,26]. Bringing this system into laboratory incubation experiments, Fukasawa and Matsukura observed a negative fungal species diversity–decomposition relationship in experiments using wood and fungi in the late stages of wood decomposition, but not in experiments using those in the early stages [27]. Hence, some relationships may exist between such a change in preference for lignocellulose decomposition and the negative species diversity–decomposition relationship during the wood decay process. Considering that the fungal interspecific interaction in wood is dominated by competition for space [11], species-rich communities within a known volume inherit a larger area of species interaction than species-poor communities. It is vital to elucidate the relationship among fungal interspecific interaction, magnitude of interaction area, and fungal lignocellulose decomposition to better understand the fungal wood decomposition process.

The purpose of this case study is to examine the effects of wood surface area on decomposition by interacting white rot basidiomycetes dominating in late stages of *P. densiflora* wood decomposition using laboratory incubation experiments. We prepared two types of wood blocks with different surface areas but equal volume and mass to determine the effects of surface area, irrespective of volume. These wood blocks were inoculated and occupied by white rot basidiomycete species. The colonized wood blocks were placed onto agar plates colonized by the same or other basidiomycete species so that the interaction between the fungus inside the wood and the fungus on the agar plate occurs on the surface of the blocks.

## 2. Materials and Methods

### 2.1. Preparation of Colonized Wood Blocks

Kiln-dried pine (*P. densiflora*) sapwood was cut into blocks of two distinct geometries (Table 1), viz., 2 × 2 × 2 cm (cube) and 0.5 × 4 × 4 cm (flat). These wood blocks have the same volume, and their dry weights were not significantly different from each other (Appendix A), whereas flat blocks have a surface area 1.7 times larger than that of cube blocks. All wood blocks were dried at 70 °C to constant weight and weighed, labeled, soaked overnight in distilled H_2_O, and then autoclaved at 121 °C for 20 min in double, sealed autoclave bags. Autoclaving was repeated three times with 1-day intervals [28]. Sterilized wood blocks were placed directly on mycelium of four white rot basidiomycetes (Table 2) grown on 0.5% malt extract agar (MA: 5 g L^−1^ malt extract, 15 g L^−1^ agar; Nakalai Tesque, Kyoto, Japan) in non-vented Petri dishes (2.5 cm thick, 14 cm in diameter), with three cubes and three flats per dish (Figure 1). All fungal strains were originally isolated from *P. densiflora* dead wood. These are late successional cord-forming species dominating late stages of wood decomposition [29]. *Phanerochaete velutina* is known as a lignin-selective white rot fungus and *Resinicium bicolor* is known as a simultaneous white rot fungus degrading lignin and carbohydrates simultaneously [29]. The other two species also belong to genera known as white rot fungi, but their preferences for lignin are not known. The dishes were sealed with Parafilm (Bemis Company Inc., Neenah, WI, USA) and incubated at 25 °C in the dark for 4 months before use in competition experiments. Nine blocks for each cube and flat for each fungus were harvested after 3 months of incubation to compare the decay abilities of each fungus in pure culture before competition. We prepared a total of 48 inoculated blocks (2 block types × 4 competitors × 6 replicates) for competition experiments, including self-pairings (pure cultures), and 18 blocks (2 block types × 9 replicates) for 3-month incubations for each fungal species.

### 2.2. Competition Experiment

After the 4-month incubation period, the pre-colonized cube and flat wood blocks, scraped free of surface mycelia and excess agar, were placed again on the cultures of four white rot basidiomycetes grown on 0.5% MA plates (Figure 1). The combinations of fungal strains in the wood and plate followed the full-factorial design, including combinations of same strains in both wood and plate (i.e., self-pairings). The dishes were sealed with Parafilm and incubated at 25 °C in the dark for an additional 4 months (i.e., a total of 8-month incubation). Three cubes and three flats were placed in a dish. The outcomes of competition in the dishes were recorded at 1, 2, and 4 (on harvest) months after the competition setup. If fungal hyphae grew out from the wood blocks onto the agar, it was recorded as a “win (replacement)” of the strains in the blocks. However, if fungal hyphae grew up onto the wood blocks from the agar media, it was recorded as a “loss (replaced)” of the strains in the blocks. We did not observe “deadlock” outcomes where neither of the fungal species grew onto the other. In addition to the dishes with fungi, we prepared 12 sterile control blocks for each cube and flat. After the incubation period, the wood blocks were harvested, dried at 70 °C, and weighed. The percentage mass loss after incubation of the original weight was calculated for each wood block as follows:Weight loss (%) = (original weight − harvest weight)/original weight × 100(1)

The average weight loss of 12 control wood blocks was subtracted from the harvest weight.

### 2.3. Acid-Unhydrolyzable Residue (Klason Lignin) Analysis

Dried wood blocks were ground in a laboratory mill to pass through a 0.5-mm screen and then used for the quantification of acid-unhydrolyzable residue (AUR, also known as the acid-insoluble residue or Klason lignin fraction). The amount of AUR in the sample was estimated gravimetrically by hot sulfuric acid digestion [30]. Samples were extracted using alcohol–benzene at room temperature, and the residues were treated with 72% sulfuric acid (*v*/*v*) for 2 h at room temperature with occasional stirring. The mixture was then diluted with distilled water to make a 2.5% sulfuric acid solution and autoclaved at 120 °C for 60 min. After cooling, the residue was filtered and washed with distilled water through a porous crucible (G4), dried at 105 °C, and then weighed. Previously, through nuclear magnetic resonance analysis, Preston et al. [31] demonstrated that the organic residue remaining after the acid hydrolysis of a variety of litter types contained a mixture of organic compounds, including not only lignin but also condensed tannins and waxes. In the present study, we use the term lignin to refer to the organic residue remaining after the acid hydrolysis of the samples for the sake of simplicity. Lignin loss relative to wood weight loss (L/W) was calculated as an index of lignin selectivity.

### 2.4. Data Analysis

Weight loss (%) and L/W of wood blocks were compared among the four fungal strains in pure culture using the post-hoc Nemenyi test, which was also applied for comparison of the weight loss and L/W of wood blocks colonized by certain species among the four competition experiments (including self-pairings). The comparison of weight loss and L/W between flat and cube wood blocks was performed using the Wilcoxon rank sum test in each fungal combination. All statistical analyses were conducted using R version 4.0.5 [32].

## 3. Results

After the 3-month decomposition trials in pure culture, all fungal species exhibited more than doubled weight loss percentages in flat wood blocks (mean = 8.05%) compared to that in cube wood blocks (mean = 3.38%), and the differences between block types were significant in all species (Appendix A; *p* < 0.001, Wilcoxon rank sum test). The weight loss (%) of cube blocks was significantly larger in Pb and Rb than in So, but it was not significantly different among Pb, Pv, and Rb. The weight loss (%) of flat blocks was not significantly different among all strains.

After the 8-month decomposition trials in pure culture, the flat blocks again exhibited more than doubled weight loss percentages (mean = 15.98%) compared with cube blocks (mean = 6.94%) (Figure 2A). In cube blocks, the weight loss (%) of 8-month pure cultured blocks (self-pairings) was not significantly different among Pb, Pv, and Rb, whereas that of So was significantly smaller than that of Pb and Rb (Nemenyi test, *p* < 0.05). In flat blocks, there was no significant difference in the weight loss (%) of 8-month pure cultured blocks among the four fungal strains. L/W was not significantly different among the four fungal strains in both block types (Figure 2B). Block type had no significant impact on L/W, except for Rb, where the L/W of cube was significantly smaller than that of flat.

The competition outcomes after the 4-month competition are shown in Table 3. Both types of wood blocks demonstrated almost similar competition outcomes in each pairing, except for Rb blocks on Pv media, where Rb cubes won, but Rb flats lost. Among the four fungal strains, Rb was the strongest competitor as it won all three competitions in cubes, although it lost one competition in flats. Pv was the second strongest as it recorded wins in two of three competitions in both block types, followed by So, which lost two competitions in both cubes and flats. Pb was the weakest competitor as it lost all three competitions in both block types.

The four-month competition after the four-month pure culture affected the weight loss of wood blocks depending on the species (Figure 3). Pb cube blocks exhibited significantly larger weight losses on Pb medium (self-pairing) and Rb medium than on So medium. Pv cube blocks on Rb medium exhibited significantly larger weight loss than that on So medium. Cube blocks of So exhibited significantly larger weight loss on Rb medium than on So medium (self-pairing). In flat blocks, competition with other fungal species exerted significant effects on weight loss in Pv, Rb, and So blocks. The weight loss of Pv flat blocks was greater on Rb medium than on Pv medium (self-pairing). The weight loss of Rb flat blocks was greater on Pv medium than on Rb medium (self-pairing). The weight loss of So flat blocks was greater on Pv medium than on So medium (self-pairing). The L/W of wood blocks was not significantly different among the competitor species in the media, except for Pv cube blocks (Figure 4). The Pv cube blocks on So medium exhibited significantly larger L/W than those on Rb medium. The cube blocks exhibited significantly lower L/W than did flat blocks in some pairings (Pv cubes on Pb and Rb media and Rb cubes on Rb and So media).

## 4. Discussion

The present study demonstrated that weight loss (%) was larger in flat wood blocks than in cube wood blocks. Considering that the initial weights of cube and flat blocks were not significantly different from each other (Appendix A), the difference in weight loss (%) directly reflected their absolute weight loss. This is probably due to the difference in the efficiency of material exchange between the interior and exterior of wood blocks. The larger surface area of flat blocks enables its exchange more efficiently than that of cube blocks. We observed some pigmentation in agar media beneath wood blocks in competition and pure-culture experiments after the harvest, indicating leakages from wood blocks during incubation. We did not observe any pigments in the agar beneath the control wood block without fungi. Although chemical analyses are required, these pigments may be diffusible phenolics released owing to lignin decomposition by white rot fungi used in the experiments. These results indicate that leaching of decay products from the wood block is vital for its decay process. However, instead of fourfold difference between flat and cube blocks in their bottom surface that contacted to the media, the difference in weight loss was approximately twofold, which is rather closer to the difference in surface area between flat and cube blocks (Table 1). Hence, gaseous exchange efficiency, particularly CO_2_ efflux from wood surface due to its decomposition [33], may be important for the weight loss of wood blocks. Chambers et al. [34] reported that respiratory emissions to the atmosphere from dead wood were predicted to be 65–88% of the total carbon loss. The larger surface area of flat blocks improves gaseous regimes in the blocks, which also merit fungal decay activities within wood. Interestingly, a significant difference in weight loss among the four fungal species in pure culture was observed only in cube and not in flat wood blocks. Such difference between cube and flat blocks may also reflect their difference in gaseous exchange efficiency, i.e., good exchange in flat blocks may cause less CO_2_ stress within the blocks, but less exchange in cube blocks may cause more CO_2_ stress. Hintikka and Korhonen [35] reported higher CO_2_ concentrations within woody litters (30% CO_2_, 5% O_2_) than those in the atmosphere (0.03% CO_2_, 21% O_2_). Adaptations to high CO_2_ conditions are highly varied among wood decomposer basidiomycetes. Chapela et al. [36] reported that the activities of cord-forming basidiomycetes, including species used in the present study, were largely reduced under high CO_2_. Low oxygen availability also retards lignin decomposition by white rot fungi because it includes oxidation reactions [37,38]. The L/W of the four fungal species was in the range of white rot fungi [39] but was smaller in cubes than in flats (particularly significant in Rb blocks), indicating that the fungi shifted their decay preferences to carbohydrates from lignin in cube blocks, probably due to low oxygen conditions in the cube center, even though *P. velutina* is known to be a selective lignin decomposer [29]. Although other researchers have discussed the effect of wood size on decomposition via water conductivity [40], this may not be the case in the present study because the wood sizes were too small to cause substantial differences in water permeation between cube and flat blocks.

In competition experiments, the effects of competitor species on decomposition were recorded in both wood types (Figure 3). However, the effect of fungal interactions on wood decomposition was dubious in cube blocks because the weight loss (%) of cube blocks was significantly different among the fungal species in pure culture (Figure 2). All significant differences in cube weight loss (%) in the competition experiment were due to reduced weight loss (%) on So medium. Given that the cube decay rate of So was significantly smaller than that of Pb and Rb, it was uncertain whether the reduced weight loss (%) on So medium was due to the fungal interaction or simply due to the replacement of original wood colonizers by the competitors in the medium. In fact, Pb cubes were replaced by So (Table 2), suggesting that the reduced weight loss (%) of Pb cubes on So medium compared to that on Pb medium (self-pairing) could be due to decomposition by the invader (So). Similarly, So cubes were replaced by Rb (Table 2), suggesting that the increased weight loss (%) of So cubes on Rb medium compared to that on So medium (self-pairing) could be due to decomposition by the invader (Rb). Moreover, Pv cubes were replaced by Rb and So (Table 2), suggesting that the difference in weight loss (%) between the Pv cubes on Rb and So media could reflect the cube decay abilities of Rb and So, and not Pv.

In flat wood blocks, the increased weight loss (%) compared with pure culture recorded in Pv, Rb, and So was the effect of competition on wood decomposition because the original decay rate of the four fungal species was not different from each other for flat blocks (Figure 2). These results partially supported our hypothesis that wood with larger surface (flat) might exhibit larger effect than wood with smaller surface (cube). However, in contrast to our prediction, the effect was positive. In particular, the pairing of Pv and Rb increased the weight loss of flat blocks irrespective of their positions (wood blocks/agar medium). There are two possible explanations for the accelerated decomposition with fungal species interaction, i.e., (1) complementary decomposition of wood components and (2) decay acceleration to compensate carbon cost for competition. Complementary decomposition of wood components is known in the pairs of cellulose decomposers and sugar fungi that utilize only low-molecular-weight sugars. The negative feedback of cellulase enzyme production by cellulose decomposers due to the high concentration of sugars produced by themselves could be reduced by sugar utilization by sugar fungi [41]. Moreover, delignification by white rot fungi also increases the wood decay abilities of cellulose decomposers without lignin decay abilities [42,43]. However, this is not the case in the present study because all the four species are white rot fungi and have similar decay preferences for wood components as shown in Figure 2.

Compensatory decay acceleration might be more likely in the present study. Fukasawa et al. [20] reported accelerated decomposition in the competition of white rot basidiomycetes particularly in small wood blocks. Similarly, several previous studies have reported accelerated weight loss of wood and CO_2_ emission in the competition between wood decomposer basidiomycetes [13,14,15,44,45,46]. In the present study, the L/W of Pv cube was significantly smaller on Rb medium than on So medium (Figure 4), although the L/W of pure culture showed no difference among the four species (Figure 2). This is probably due to the shift of decay preference for carbohydrates to compensate carbon cost for competition rather than resource partitioning between fungal species. Chi et al. [47] also reported that co-cultures of the lignin-selective white rot fungi *Ceriporiopsis subvermispora*, *Physisporinus rivulosus*, *Phanerochaete chrysosporium*, and *Pleurotus ostreatus* suppressed lignin degradation compared to their monocultures and promoted the decomposition of hemicelluloses. It has been reported that even selective lignin decomposers increase carbohydrate decomposition compared with lignin decomposition under stressful conditions such as drought [48]. Although some upregulations of lignin-degrading enzymes, such as laccase, manganese peroxidase, and lignin peroxidase, were recorded in the interaction front of fungal colonies [49,50], it is likely that these enzymes are used for the production of pigments and removal of active oxygen species generated because of physiological stress rather than lignin decomposition [51].

To summarize, our results demonstrated clearly positive effects of fungal interaction on wood decomposition in flat blocks but not in cube blocks, probably because of the increased carbon demands to compensate competition cost. In contrast to our prediction, the fungal combinations used in the present study did not exert negative effects of fungal interactions on wood decomposition. This may be partially attributable to the traits of fungal species used in the present study. As mentioned, all the four species are cord-forming basidiomycetes and possess strong combative abilities [11]. All the fungal combinations used in the present study resulted in either win or loss of competitors, and no deadlock was observed (Table 3). Lack of settled interaction frontline may reduce negative effects on decomposition because the winner can actively decay wood without constant allocation to defense. Easily available carbohydrates (malt extract) in the agar may also reduce the negative effects of competition on decomposition because they can reduce the trade-offs of carbon resources in fungal mycelia. Furthermore, all the four species did not produce obvious pigments at the interaction front, which may also reduce the negative effects on decomposition because fungal melanin is often detected as acid-unhydrolyzable residue in Klason lignin analysis [19]. However, the reduced lignin loss in cube blocks under fungal interactions suggests that lignin concentration was increased even during wood decomposition by the communities of those white rot fungi, consistent with the field data [21,52]. Recalcitrant lignin is a precursor of soil organic matter that is essential for carbon sequestration [53]. Regarding lignin accumulation, it is also necessary to consider interactions with brown rot fungi, which degrade only the wood carbohydrates, celluloses and hemicelluloses. Although previous studies have reported that the effects of interspecific interactions on wood decay rate were not obviously different between white rot and brown rot fungi and there are no clear compensatory decomposition synergistic effects between them [16,20], their interactions exert significant effects on lignocellulose decomposition [54]. Future studies should include a larger set of species with different decay types (white rot, brown rot, and soft rot) to evaluate their interactions and effects on wood decomposition. The wood of broadleaf trees should also be used because wood species affects the outcome of fungal competition and decomposition [16].

## Figures and Tables

**Figure 1 jof-08-00517-f001:**
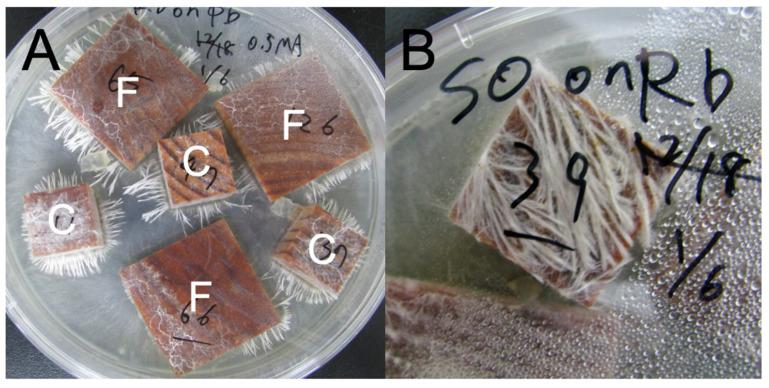
(**A**) Incubation of cube (C) and flat (F) wood blocks colonized by *Resinicium bicolor* (Rb) on *Pholiota brunnescence* (Pb) colony growing on 0.5% malt extract agar. (**B**) Cube wood blocks colonized by *Scytinostroma odoratum* (So) on Rb colony. Note that Rb hyphae grew out from wood onto the agar in (**A**), indicating a “win” of Rb, but Rb hyphae grew from agar onto wood blocks in (**B**), indicating a “loss” of So.

**Figure 2 jof-08-00517-f002:**
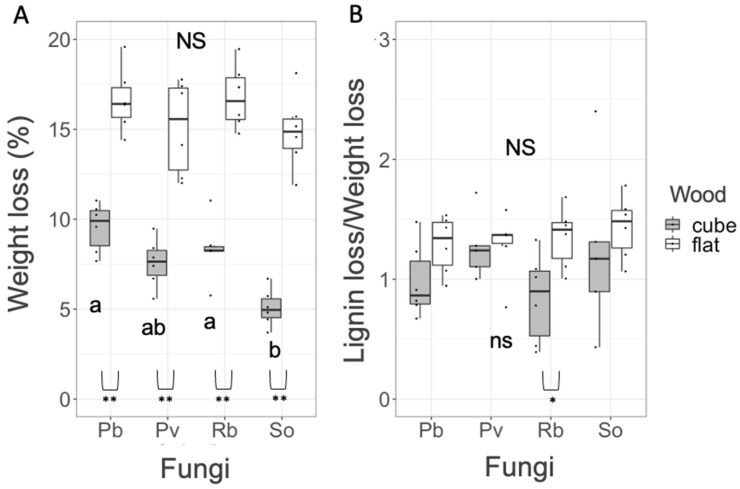
Weight loss (**A**) and lignin loss/weight loss (**B**) of wood blocks after 8-month incubation in pure culture of the four fungal species. NS and the same alphabets indicate no significant differences among fungal species in the post hoc Nemenyi test (uppercase, flat; lowercase, cube). Comparison between flat and cube blocks within a species was performed using the Wilcoxon rank sum test (*, *p* < 0.05; **, *p* < 0.01). *n* = 6.

**Figure 3 jof-08-00517-f003:**
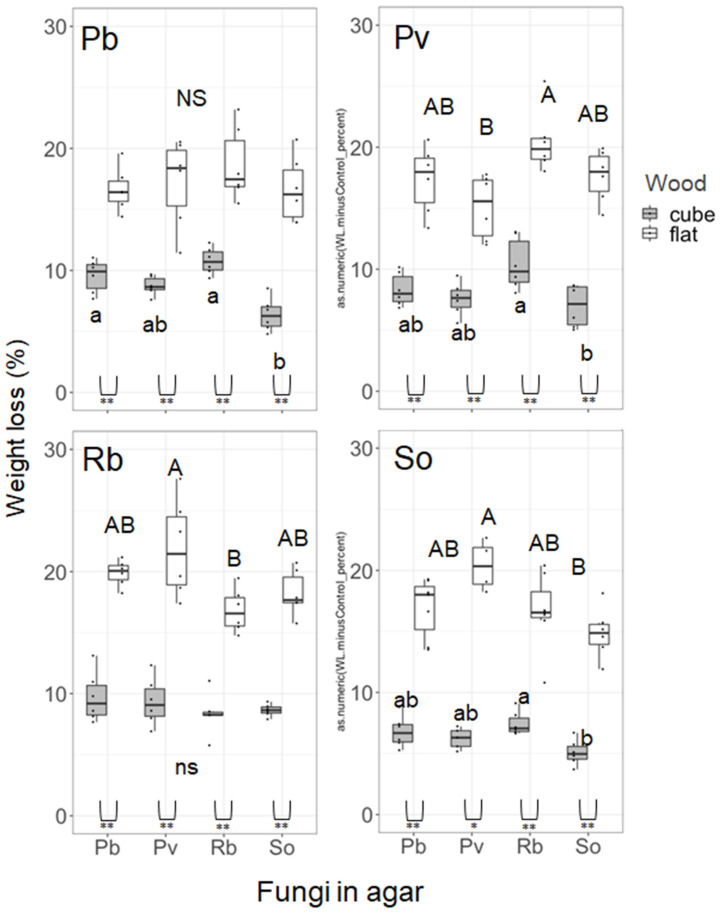
Weight loss of wood blocks after 4-month incubation in pure culture + 4-month incubation in competition of the four fungal species. NS and the same alphabets indicate no significant differences among fungal species in the post hoc Nemenyi test (uppercase, flat; lowercase, cube). Comparison between flat and cube blocks within a species was performed using the Wilcoxon rank sum test (*, *p* < 0.05; **, *p* < 0.01). *n* = 6.

**Figure 4 jof-08-00517-f004:**
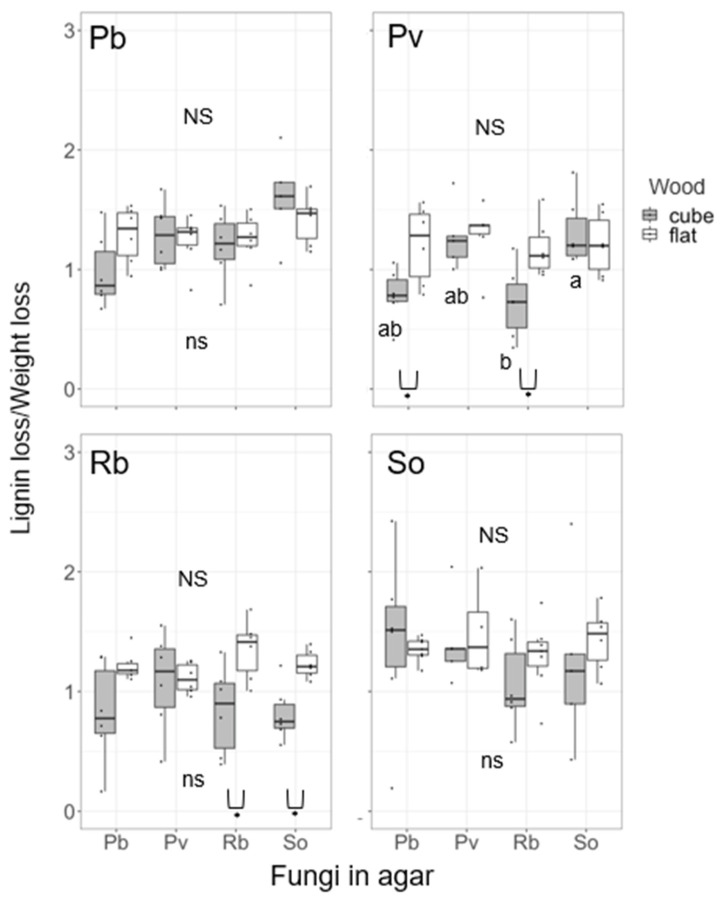
Lignin loss/weight loss of wood blocks after 4-month incubation in pure culture + 4-month incubation in competition of the four fungal species. NS and the same alphabets indicate no significant differences among fungal species in the post hoc Nemenyi test (uppercase, flat; lowercase, cube). Comparison between flat and cube blocks within a species was performed using the Wilcoxon rank sum test (*, *p* < 0.05). *n* = 6.

**Table 1 jof-08-00517-t001:** Geometry of pine wood blocks used in the present study.

Geometry	Cube	Flat
Surface area (cm^2^)	24	40
Volume (cm^3^)	8	8
Surface area/volume	3	5
Agar contact (cm^2^)	4	16

**Table 2 jof-08-00517-t002:** Fungal species used in the present study.

Fungi	Abbreviation	NBRC * Code
*Pholiota brunnescens* A.H. Sm. & Hesler	Pb	110175
*Phanerochaete velutina* (DC.) P. Karst.	Pv	110184
*Resinicium bicolor* (Alb. & Schwein.) Parmasto	Rb	110186
*Scytinostroma odoratum* (Fr.) Donk	So	110188

* NITE Biological Resource Center.

**Table 3 jof-08-00517-t003:** Competition outcome of fungi in wood blocks against fungi in the agar media.

Agar	Pb-Cube	Pv-Cube	Rb-Cube	So-Cube	Pb-Flat	Pv-Flat	Rb-Flat	So-Flat
Pb	–	win	win	win	–	win	win	win
Pv	loss	–	win	loss	loss	–	loss	loss
Rb	loss	loss	–	loss	loss	loss	–	loss
So	loss	win/loss	win	–	loss	win/loss	win	–

Win, fungi in wood blocks grew out into the agar media. Loss, fungi in the agar media grew up onto the wood blocks. Note that the results were obtained from qualitative observations of six replicates for each combination.

## Data Availability

The data presented in this study are available on request from the corresponding author.

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
