# Peer review of "Surface Area of Wood Influences the Effects of Fungal Interspecific Interaction on Wood Decomposition—A Case Study Based on *Pinus densiflora* and Selected White Rot Fungi"

_jof, 2022, doi:10.3390/jof8050517_

Round 1
Reviewer 1 Report
„Surface-to-volume ratio of wood influences the effects of fungal interspecific interaction on wood decomposition” is an interesting paper aimed at evaluating the effect of wood dimension on fungal interactions and the resulting effectiveness of wood decomposition. However, I have some concerns, question and suggestions:
- Since the study involves only four white-rot fungi species and one wood species, I strongly suggest to change the title and specify there that the research concerns only selected white-rot and Pinus densiflora (perhaps adding “ – a case study based on Pinus densiflora and selected white-rot fungi” at the end of title will be good); I also suggest to change the perspective for the whole manuscript and write is as a case study.
- Since this is a case study, based on selected fugal and wood species and focusing only on surface to volume ratio of wood, please re-write the conclusions made upon the research so they reflect the real scope of experiments performed. Also, please explain clearly in the manuscript why the surface to volume ratio of wood and weight and lignin loss of wood were selected as factors of importance in your study. Why not the production of secondary metabolites, that may be harmful to different fungi or limit their activity, or hyphal competition, variation in sets of enzymes produced by different fungi species?
- Please elaborate on the importance of the research. Are the results useful in practice? If so, add some information about that.
Author Response
Please see my response in attached word file.

Reviewer 2 Report
Main question is: Why did you combine a softwood species with white-rot fungi and did not use either a hardwood for white-rot fungi or your pine wood for brown-rot fungi? Future experiments should consider this. Also a combination of both groups would be interesting.
You should also clearly consider that there are different groups of white-rot fungi, simultaneous and selective ones, and add to your paper to which group your used species belong.
I made some remarks as commentaries in the attached reviewed pdf version of your manuscript.

Author Response

(The authors gave the same response as above.)

Round 2
Reviewer 1 Report
The manuscript has been thoroughly corrected. I recommend it for publishing.
This manuscript is a resubmission of an earlier submission. The following is a list of the peer review reports and author responses from that submission.
Round 1
Reviewer 1 Report
„Surface-to-volume ratio of wood influences the effects of fungal interspecific interaction on wood decomposition” is an interesting paper to evaluate the effect of wood dimension on fungal interactions and the effectiveness of wood decomposition. I have some detailed comments, questions and suggestions on the manuscript – they can be found in the pdf file attached.

Reviewer 2 Report
In general, the topic of this manuscript is scientifically sound and valuable. However, in my opinion, the experimental design of this manuscript couldn’t support the topic and the results were simple. The influencing mechanism couldn’t be explained too. Some major problems are shown as followed:
- The surface area/volume ratio isn’t enough. Only one pair couldn’t represent the surface-to-volume ratio of wood, especially it was the major influence factor.
- In this experiment, only pine wood, a kind of gymnosperm, was tested. In fact, angiosperm wood are more than gymnosperm in nature forest.
- Fungal interspecific interaction include several contents, hyphal competition, secondary metabolites production and enzyme activity variation. The effects of interaction should be more than the weight and lignin loss of wood. From the topic, we look forward to know above mentioned results during two species competing with different SA/V ratio.
- It is one-sided that only four white rot species were used in this experiment. However, brown rot fungi are also important during wood decomposition with different mechanism completely. Perhaps they will take different results.
In sum, the results of this experiment based on one tree species wood, one pair of SA/V ratio, fewer physicochemical property tested couldn’t demonstrate the influence the effects of fungal interspecific interaction. The author need to design the experiment with more directions and details.